# ADAPTIVEGAUSSIAN: GENERALIZABLE 3D GAUSSIAN RECONSTRUCTION FROM ARBITRARY VIEWS

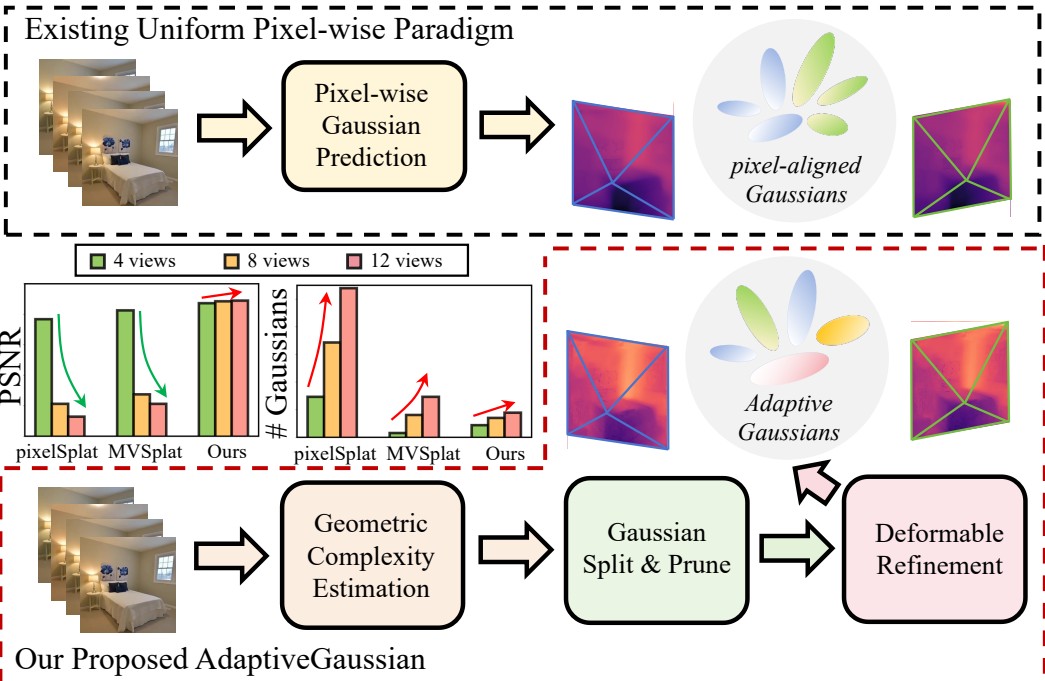

Figure 1: Most existing generalizable 3D Gaussian splatting methods (e.g., pixelSplat (Charatan et al., 2023), MVSplat (Chen et al., 2024)) assigns a fixed number of Gaussians to each pixel, leading to inefficiency in capturing local geometry and overlap across views. Differently, our AdaptiveGaussian dynamically adjusts the Gaussian distributions based on geometric complexity in a feed-forward framework. With comparable efficiency, AdaptiveGaussian (trained using 2 views) successfully generalizes to various numbers of input views with adaptive Gaussian densities.

## ABSTRACT

We propose **AdaptiveGaussian**, an efficient feed-forward framework for learning generalizable 3D Gaussian reconstruction from arbitrary views. Most existing methods rely on uniform pixel-wise Gaussian representations, which learn a fixed number of 3D Gaussians for each view and cannot generalize well to more input views. Differently, our AdaptiveGaussian dynamically adapts both the Gaussian distribution and quantity based on geometric complexity, leading to more efficient representations and significant improvements in reconstruction quality. Specifically, we introduce a Cascade Gaussian Adapter (CGA) to adjust Gaussian distribution according to local geometry complexity identified by a keypoint scorer. CGA leverages deformable attention in context-aware hypernetworks to guide Gaussian pruning and splitting, ensuring accurate representation in complex regions while reducing redundancy. Furthermore, we design a transformer-based Iterative Gaussian Refiner (IGR) module that refines Gaussian representations through direct image-Gaussian interactions. Our AdaptiveGaussian can effectively reduce Gaussian redundancy as input views increase. We conduct extensive experiments on the large-scale ACID and RealEstate10K datasets, where

our method achieves state-of-the-art performance with good generalization to various numbers of views.

# 1 INTRODUCTION

Novel view synthesis (NVS) seeks to reconstruct a 3D scene from a series of input views and generate high-quality images from previously unseen viewpoints. High-quality and real-time reconstruction and view synthesis are crucial for autonomous driving (Tonderski et al., 2023; Khan et al., 2024; Tian et al., 2024), robotics perception ( Wilder-Smith et al., 2024; Jiang et al., 2023a) and virtual or augmented reality ( Yang et al., 2024; Zheng et al., 2024).

NeRF-based methods ( Mildenhall et al., 2020; Hu et al., 2022; Liu et al., 2020; Neff et al., 2021) have achieved remarkable success by encoding 3D scenes into implicit radiance fields, yet sampling volumes for NeRF rendering is costly in both time and memory. Recently, Kerbl et al. (2023) proposed to represent 3D scenes explicitly using a set of 3D Gaussians, enabling much more efficient and high-quality rendering via a differentiable rasterizer. Still, the original 3D Gaussian Splatting requires separate optimization on each single scene, which significantly reduces inference efficiency. To tackle this problem, recent researches have aimed at generating 3D Gaussians directly from a feed-forward network without any per-scene optimization ( Charatan et al., 2023; Chen et al., 2024; Liu et al., 2024; Szymanowicz et al., 2024; Zheng et al., 2024). Typically, these approaches adhere to a paradigm where a fixed number of Gaussians is predicted for each pixel in the input views. The Gaussians derived from different views are then directly merged to construct the final 3D scene representation. However, such a paradigm limits the model performance as the Gaussian splats are uniformly distributed across images, making it difficult to capture local geometric details effectively. Additionally, as the number of input views increases, directly merging Gaussians can degrade reconstruction performance due to severe Gaussian overlap and redundancy across views.

To address this, we propose **AdaptiveGaussian**, which enables dynamic adaption on both 3D Gaussian distribution and quantity. To be specific, we first uniformly initialize Gaussian positions following Chen et al. (2024) to accurately localize the Gaussian centers. To identify geometry complexity across images, we then compute a relevance score map for each input view from image features in an end-to-end manner. Under the guidance of score maps, we construct a Cascade Gaussian Adapter (CGA), which leverages deformable attention (Xia et al., 2022) to control the pruning and splitting operations. After CGA, more Gaussians are allocated to regions with complex geometry for precise reconstruction, while unnecessary and duplicate Gaussians across views are pruned to reduce redundancy and improve efficiency. Since these Gaussian representations still fall short in fully capturing the image details, we further introduce a transformer-based Iterative Gaussian Refiner (IGR) to refine 3D Gaussians through direct image-Gaussian interactions. Finally, we employ rasterization-based rendering using the refined Gaussians to generate novel views at target viewpoints.

We conduct extensive experiments on ACID (Liu et al., 2021a) and RealEstate10K (Zhou et al., 2018) benchmarks for large-scale 3D scene reconstruction and NVS. AdaptiveGaussian outperforms existing methods on arbitrary input views with a comparable inference speed. Notably, compared to previous pixel-wise methods which generate uniform pixel-aligned Gaussian predictions, our model mitigates Gaussian overlap and redundancy across views by dynamically adjusting their distribution based on local geometry complexity, leading to much more precise reconstruction as the number of input views increases, achieving a PSNR improvement of around 6 dB compared to pixel-wise methods. Visualizations and ablations further demonstrate that both CGA and IGR blocks are crucial in adapting Gaussian distribution, capturing geometry details, and improving reconstruction accuracy.

# 2 RELATED WORK

**Multi-View Stereo.** Multi-View Stereo (MVS) aims to reconstruct a 3D representation from multi-view images of a given scene or object. Since accurate depth estimation is essential for reliable 3D reconstruction from 2D inputs, most MVS methods ( Gu et al., 2020; Ding et al., 2021; Yao et al., 2018) require ground truth depth for supervision in training process. Additionally, point-based MVS approaches generally separate the processes of depth estimation and point cloud fusion processes. Recently, inspired by efficient Gaussian representations proposed by Kerbl et al. (2023),

Chen et al. (2024) introduces to directly predict depth for pixel-wise Gaussians from a cost volume structure without requiring depth supervision, significantly improving model scalability and flexibility. Therefore, following a similar approach, we construct a lightweight cost volume to facilitate depth estimation, which serves as an efficient initialization for 3D Gaussians in our AdaptiveGaussian.

**Per-scene 3D Reconstruction.** Neural Radiance Fields (NeRF) have revolutionized the field of 3D reconstruction by representing scenes as implicit neural fields (Mildenhall et al., 2020). Subsequent researches have focused on overcoming the limitations of the original NeRF to improve its performance and broaden its applicability. Some researches aim to improve the efficiency for novel view synthesis ( Hu et al., 2022; Fridovich-Keil et al., 2022; Yu et al., 2021a; Liu et al., 2020; Neff et al., 2021). Moreover, several studies concentrate on capturing intricate geometry and temporal information to achieve accurate and dynamic reconstruction ( Li et al., 2021; Du et al., 2021; Pumarola et al., 2020; Tian et al., 2023; Wang et al., 2022). Compared to implicit NeRF-based methods, 3D Gaussian Splatting (3DGS) (Kerbl et al., 2023) represents a 3D scenario as a set of explicit 3D Gaussians, enabling a rasterization-based splatting rendering process that is significantly more efficient in both time and memory. Given that 3DGS still requires millions of 3D Gaussians to represent a single scene, numerous studies have focused on achieving real-time rendering and minimizing memory usage ( Fan et al., 2023; Katsumata et al., 2024; Lu et al., 2024). Additionally, some researches focus on enhancing the reconstruction quality of 3DGS by employing multi-scale rendering (Yan et al., 2024), advanced shading models (Jiang et al., 2023b) or incorporating physically based properties for realistic relighting (Gao et al., 2023). However, these methods still require per-scene optimization and rely on dense input views, which can be computationally expensive and limit their scalability for large-scale or dynamic scenes.

**Generalizable 3D Reconstruction.** PixelNeRF (Yu et al., 2021b) pioneers the approach of predicting pixel-wise features directly from input views to reconstruct neural radiance fields. Following methods incorporate volume or transformer architectures to improve the performance of feed-forward NeRF models ( Chen et al., 2021a; Xu et al., 2024; Miyato et al., 2024; Sajjadi et al., 2022; Du et al., 2023). However, these feed-forward NeRF approaches typically demand substantial memory and computational resources due to the expensive per-pixel volume sampling process (Wang et al., 2021a; Johari et al., 2022; Barron et al., 2021; Garbin et al., 2021; Reiser et al., 2021; Müller et al., 2022). With the advent of 3DGS, PixelSplat (Charatan et al., 2023) initiates a shift towards feed-forward Gaussian-based reconstruction. It takes sparse input views to directly predict pixel-wise 3D Gaussians by leveraging epipolar geometry to learn cross-view features. MVSplat (Chen et al., 2024) constructs a cost volume structure for depth estimation, which significantly boosts both model efficiency and reconstruction quality. Additionally, MVSGaussian (Liu et al., 2024) further improves model performance by introducing an efficient hybrid Gaussian rendering process. Moreover, SplatterImage (Szymanowicz et al., 2024) and GPS-Gaussian (Zheng et al., 2024) predict pixel-wise 3D Gaussians for object-centric or human reconstruction.

However, these feed-forward methods are constrained by the pixel-wise Gaussian prediction paradigm, which limits the model's performance as the Gaussian splats are uniformly distributed across images. Such a paradigm inadequately captures intricate geometries, while also causing Gaussian overlap and redundancy across views, ultimately resulting in severe rendering artifacts. In comparison, AdaptiveGaussian consists of a Cascade Gaussian Adapter (CGA), allowing for dynamic adaption on both Gaussian distribution and quantity. Visualizations demonstrate that CGA is capable of allocating more Gaussians in areas rich in geometric details, while reducing duplicate Gaussians in similar regions across input views. Furthermore, we introduce an Iterative Gaussian Refiner (IGR), enabling direct interaction between 3D Gaussians and local image features via deformable attention. Experimental results show that IGR effectively leverages image features to guide Gaussians in capturing the full information contained within the images, significantly enhancing the model's ability to capture local intricate geometry.

## 3 PROPOSED APPROACH

In this section, we present our method to learn generalizable Gaussian representations from arbitrary views. Given an arbitrary set of input images $\mathcal{I} = \{I_i\}_{i=1}^{N} \in \mathbb{R}^{N \times H \times W \times 3}$ and corresponding camera poses $\mathcal{C} = \{C_i\}_{i=1}^{N}$, our AdaptiveGaussian aims to learn a mapping $\mathcal{M}$ from images to 3D

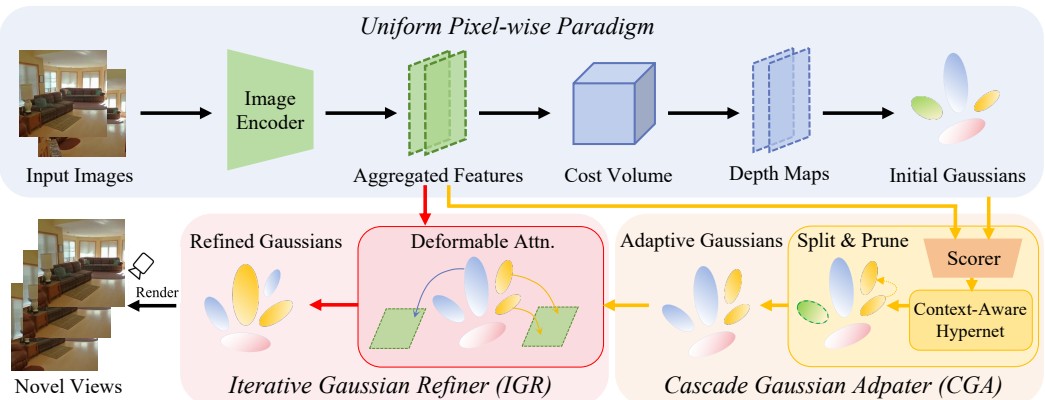

Figure 2: **Overview of AdaptiveGaussian.** Given multi-view input images, we initialize 3D Gaussians using a lightweight image encoder and cost volume. Cascade Gaussian Adapter (CGA) then dynamically adapts both the distribution and quantity of Gaussians. By leveraging local image features, Iterative Gaussian Refiner (IGR) further refines Gaussian representations via deformable attention. Finally, novel views are rendered from the refined 3D Gaussians using rasterization-based rendering.

Gaussians for scene reconstruction:

$$\mathcal{M} : \{(I_i, C_i)\}_{i=1}^{N} \mapsto \{(\mu_j, s_j, r_j, \alpha_j, sh_j)\}_{j=1}^{N_K}, \tag{1}$$

where $N_K$ is the total number of 3D Gaussians, which adaptively varies depending on the scene context. Each Gaussian is parameterized by its position $\mu_j$, scaling $s_j$, rotation $r_j$, opacity $\alpha_j$ and spherical harmonics $sh_j$.

As illustrated in Figure 2, we first use a lightweight cost volume for depth estimation and Gaussian position initialization. We then introduce Cascade Gaussian Adapter (CGA), which dynamically adapts both Gaussian distribution and quantity based on local geometric complexity. Finally, we explain how Iterative Gaussian Refiner (IGR) enables direct image-Gaussian interactions, further refining Gaussian distribution and representations for enhanced reconstruction.

## 3.1 GAUSSIAN INITIALIZATION

**Position Initialization.** Following the instructions of MVSplat (Chen et al., 2024), we first extract image features via a 2D backbone consisting of CNN and Swin Transformer (Liu et al., 2021b). Specifically, CNN encodes multi-view images to corresponding feature maps, while Swin Transformer performs both self-attention and cross-view attention to better leverage information cross views. Then, we obtain the aggregated multi-view features $\mathcal{F} = \{F_i\}_{i=1}^{N}$.

To initialize Gaussian positions precisely, we construct a lightweight cost volume (Yao et al., 2018) for depth estimation, denoted as $\Phi_{depth}$. We then predict Gaussian centers as follows:

$$\mu = P^{-1}(\Phi_{depth}(\mathcal{F}), \mathcal{C}) \tag{2}$$

where $P^{-1}(\cdot)$ stands for unprojection operation.

**Parameter Initialization.** For each Gaussian center $\mu_j$, we randomly set corresponding scaling $s_j \in \mathbb{R}^3$, rotation $r_j \in \mathbb{R}^4$, opacity $\alpha_j \in \mathbb{R}^1$, spherical harmonics $sh_j \in \mathbb{R}^C$ within a proper range. we then get the initial Gaussians set $\mathcal{G} = \{(\mu_j, s_j, r_j, \alpha_j, sh_j)\}_{j=1}^{HW} \in \mathbb{R}^{HW \times (11+C)}$.

## 3.2 CGA: CASCADE GAUSSIAN ADAPTER

After obtaining the initial Gaussian set $\mathcal{G}$, we introduce Cascade Gaussian Adapter (CGA) driven by a multi-view keypoint scorer $\Psi$, as shown in Figure 3(a). CGA contains a set of context-aware hypernetworks $\mathcal{H}$ which dynamically control and guide the following Gaussian pruning and splitting operations. This approach ensures that regions with complex geometry details are represented by a

Figure 3: **Illustration of the proposed CGA and IGR Blocks.** (a) CGA comprises a keypoint scorer followed by a series of hypernetworks that produce context-aware thresholds to guide the splitting and pruning of Gaussians. (b) IGR further facilitates direct image-Gaussian interactions, enabling Gaussian representations to capture and extract local geometric features more effectively.

greater number of Gaussians, while areas with poor geometry can be represented with fewer Gaussians. In parallel, CGA effectively removes redundant Gaussians to prevent Gaussian overlap across views. Compared to previous pixel-wise methods, which rigidly allocate a fixed number of Gaussians per pixel, our design dynamically adapts both distribution and quantity of Gaussians based on geometric complexity. This flexibility allows for a more accurate capture of local geometry and mitigates the problem of Gaussian overlap, thereby improving the overall quality of reconstruction.

Given the aggregated features $\mathcal{F}$ derived in Section 3.1, $\Psi$ computes relevance score maps $\mathcal{R} = \{R_i\}_{i=1}^N \in \mathbb{R}^{N \times H \times W}$, where each score map $R_i$ is obtained by a learnable weighted average of contributions from different views:

$$\mathcal{R} = \Psi(\mathcal{F}) = softmax\left(MLP\left(\sum_{i=1}^N \alpha_i \cdot F_i\right)\right), \quad \alpha_i = \frac{\exp(\beta_i)}{\sum_{j=1}^N \exp(\beta_j)}, \quad (3)$$

where $A = [\alpha_1, \alpha_2, \ldots, \alpha_N]^T \in \mathbb{R}^N$ represents the contribution factor of each view, and is determined by learnable parameters $\beta_i (i = 1, 2, ..., N)$.

We first introduce a set of hypernetworks $\mathcal{H} = \{H_k\}_{k=1}^K$ to generate *context-aware* thresholds. CGA is composed of $K$ stages, where each stage $H_k$ takes score maps $\mathcal{R}$ along with Gaussian set $\mathcal{G}_k = \{(\mu_j^{(k)}, s_j^{(k)}, r_j^{(k)}, \alpha_j^{(k)}, sh_j^{(k)})\}_{j=1}^{N_k} \in \mathbb{R}^{N_k \times (11+C)}$ as input, and outputs thresholds $\tau_{high}^{(k)}, \tau_{low}^{(k)} \in \mathbb{R}$ for splitting and pruning. As illustrated in equation 4, we first sample and embed Gaussian set $\mathcal{G}_k$ into Gaussian score queries $\mathcal{Q}_r^{(k)}$. Then we project sampled reference points $\mu^{(k)}$ onto score maps $\mathcal{R}$ with corresponding camera parameters $\mathcal{C}$. Finally, we update queries $\mathcal{Q}_r^{(k)}$ with weighted scores from $\mathcal{S}$ and get both thresholds through a simple MLP. Initially, we set $\mathcal{G}_1 = \mathcal{G}$.

$$\tau_{high}^{(k)}, \tau_{low}^{(k)} = \mathcal{H}_k(\mathcal{G}_k, \mathcal{R}, \mathcal{C}) = MLP(\sum_{i=1}^N \alpha_i \cdot DA(\mathcal{Q}_r^{(k)}, R_i, P(\mu^{(k)}, C_i))), \quad (4)$$

where $DA(\cdot), P(\cdot)$ denote the deformable attention function and projection operation, respectively.

Then, we obtain Gaussian-wise scores by projecting Gaussian centers onto score maps $\mathcal{R}$. To elaborate, let $S_k = \{s_{ij}^{(k)}\} \in \mathbb{R}^{N \times N_k}$ be the score matrix for Gaussian set $\mathcal{G}_k$, where each score $s_{ij}^{(k)}$ is the value at the projection point of the $j - th$ Gaussian center in $R_i$, or 0 if it does not project onto any region in $R_i$. The final Gaussian-wise scores $S_k^{avg}$ are then computed by averaging scores across different views:

$$S_k^{avg} = S_k^T \cdot A, \quad (5)$$

Once Gaussian-wise scores are obtained, regions with higher scores, indicating more complex geometry details, undergo splitting operation to allocate more Gaussians for finer representations. For regions with lower scores, we apply an opacity-based pruning operation, gradually reducing Gaussian opacity and scaling to minimize their impact and reduce redundancy.

**Splitting.** For Gaussian $g_j^{(k)} \in \mathcal{G}_k$ with score higher than $\tau_{\text{high}}^{(k)}$, we generate $M$ separate new Gaussians for more detailed representations:

$$G_j^{(k)} = SplitNet(g_j^{(k)}) \in \mathbb{R}^{M \times (11+C)}, \tag{6}$$

where $SplitNet(\cdot)$ is a simple MLP-based network that ensures all parameters within proper range. The newly generated Gaussians are then directly concatenated with the existing Gaussian set $\mathcal{G}_k$.

**Pruning.** For Gaussian $g_j^{(k)} \in \mathcal{G}_k$ with score lower than $\tau_{\text{low}}^{(k)}$, we apply an opacity-based pruning operation rather than directly removing it. Specifically, we set a predefined opacity threshold $\tau_\alpha$. If the Gaussian opacity is greater than $\tau_\alpha$, we gradually reduce its opacity and scaling:

$$\alpha_j^{(k)} \to \gamma_\alpha \cdot \alpha_j^{(k)}, \quad s_j^{(k)} \to \gamma_s \cdot s_j^{(k)}, \tag{7}$$

where $\gamma_\alpha < 1$ and $\gamma_s < 1$ are reduction factors. Otherwise, the current Gaussian is removed entirely from Gaussian set $\mathcal{G}_k$.

After K-stage adaptation in the Cascade Gaussian Adapter, the initial uniform 3D Gaussian representations are transformed into adaptive forms. Gaussians are redistributed according to geometric complexity, resulting in a more efficient and context-aware representation.

### 3.3 IGR: ITERATIVE GAUSSIAN REFINER

Though CGA allows for a more optimal Gaussian distribution, the Gaussian representations still fall short in capturing the full information contained in the images. Inspired by the efficiency demonstrated by GaussianFormer (Huang et al., 2024) in occupancy prediction, we design a transformer-based Iterative Gaussian Refiner (IGA) to further extract local geometric information from input views, as shown in Figure 3(b). In this process, we leverage deformable attention to enable direct image-Gaussian interactions, enhancing the ability for 3D Gaussians to more accurately capture intricate geometry details in reconstruction and view synthesis.

IGR is composed of $B$ attention and refinement blocks. In Section 3.2, CGA adapts the original Gaussian set $\mathcal{G}$ to $\mathcal{G} = \mathcal{G}_K$. To continue, we first sample and embed $\mathcal{G}$ into Gaussian queries $\mathcal{Q}$. In each block, deformable attention is first applied between Gaussian queries $\mathcal{Q}$ and multi-view features $\mathcal{F}$ to update Gaussian representations. This is followed by a refinement stage where a residual module further fine-tunes the queries. The overall process of IGR can be formulated as:

$$\mathcal{Q}_b = \Phi_{ref}(\sum_{i=1}^{N} \alpha_i \cdot DA(\mathcal{Q}_{b-1}, F_i, P(\mu^{(b)}, C_i))) \quad b = 1, 2, \ldots, B, \tag{8}$$

where $DA(\cdot), \Phi_{ref}(\cdot), P(\cdot)$ denote the deformable attention function, refinement layer and projection operation, $F_i, C_i, \alpha_i$ represents the image feature, camera parameters and contribution factor of input view $I_i$, respectively. $\mathcal{Q}_b(b = 1, 2, ..., B)$ stands for output queries from the $b-th$ IGR block, and $\mu^{(b)}$ is the Gaussian center of current stage. Initially, we set $\mathcal{Q}_0 = \mathcal{Q}$.

Finally, the refined Gaussian queries are decoded into Gaussian parameters $\mathcal{G}_f$ through a simple MLP to ensure all parameters within proper range, and then can be used for rasterization-based rendering at novel viewpoints.

$$\mathcal{G}_f = \{(\mu_j^f, s_j^f, r_j^f, \alpha_j^f, sh_j^f)\}_{j=1}^{N_K} = MLP(\mathcal{Q}_B). \tag{9}$$

Our full model takes ground-truth target RGB images at novel viewpoints as supervision, allowing for efficient end-to-end training. The training loss is calculated as a linear combination of MSE and LPIPS (Zhang et al., 2018) losses, with loss weights of 1 and 0.05, respectively.

Table 1: **Results of Novel View Synthesis on the RealEstate10K and ACID benchmarks.** We report the average PSNR and LPIPS (Zhang et al., 2018) on the test set, where all models are trained with 2 reference views and inferred with 4, 8 and 12 reference views.

| Datasets | Methods | 2→4 Views | | 2→8 Views | | 2→12 Views | |
|---|---|---|---|---|---|---|---|
| | | PSNR | LPIPS | PSNR | LPIPS | PSNR | LPIPS |
| RealEstate10K | pixelNeRF | 21.03 | 0.520 | 21.22 | 0.498 | 21.28 | 0.501 |
| | MuRF | 23.30 | 0.188 | 23.78 | 0.186 | 23.94 | 0.185 |
| | pixelSplat | 22.02 | 0.195 | 19.97 | 0.229 | 18.92 | 0.267 |
| | MVSplat | 22.30 | 0.185 | 20.39 | 0.216 | 19.69 | 0.233 |
| | AdaptiveGaussian | **23.95** | **0.182** | **24.05** | **0.183** | **24.18** | **0.180** |
| ACID | pixelNeRF | 20.77 | 0.508 | 21.03 | 0.487 | 21.05 | 0.485 |
| | MuRF | 25.85 | 0.193 | 26.04 | 0.190 | 26.10 | 0.191 |
| | pixelSplat | 21.08 | 0.207 | 17.70 | 0.264 | 17.30 | 0.279 |
| | MVSplat | 20.89 | 0.209 | 18.13 | 0.260 | 17.33 | 0.277 |
| | AdaptiveGaussian | **26.21** | **0.189** | **26.28** | **0.185** | **26.44** | **0.182** |

Compared to the uniform pixel-wise paradigm, our AdaptiveGaussian approach dynamically adapts both the Gaussian distribution and quantity within the Cascade Gaussian Adapter. Additionally, the Iterative Gaussian Refiner refines Gaussian representations to capture intricate geometric details in the input views. This design achieves more efficient Gaussian distributions while mitigating overlap and redundancy common in pixel-wise methods.

## 4 EXPERIMENTS

### 4.1 EXPERIMENTAL SETTINGS

**Datasets.** To assess the performance of our model, we conduct experiments on two extensive datasets: ACID (Liu et al., 2021a) and RealEstate10K (Zhou et al., 2018). The ACID dataset consists of video frames capturing natural landscape scenes, comprising 11,075 scenes in the training set and 1,972 scenes in the test set. RealEstate10K provides video frames from real estate environments, with 67,477 scenes allocated for training and 7,289 scenes reserved for testing. The model is trained with two reference views, and four novel views are selected for evaluation. During testing, we select 4, 8 and 12 views as reference views to cover as wide a view range as possible to evaluate the model performance on large-scale and wide-range scenarios.

**Implementation Details.** We set the resolutions of input images as 256x256. In Cascade Gaussian Adapter (CGA), we apply $K = 3$ stages of cascade Gaussian adaption. As for the splitting operation, the SplitNet generates $M = 1$ separate new Gaussians, whereas the pruning process uses reduction factors $\gamma_\alpha = \gamma_s = 0.5$ and opacity threshold $\tau_\alpha = 0.3$. We use $B = 3$ blocks in Iterative Gaussian Refiner (IGR) to extract local geometry from input views. We implement our AdaptiveGaussian with Pytorch and all the models are trained on a single NVIDIA A6000 GPUs for 300,000 iterations with Adam optimizer. More training details are provided in Section A.2.

### 4.2 MAIN RESULTS

**Novel View Synthesis.** As shown in Table 1 and Figure 4, our proposed AdaptiveGaussian consistently outperforms previous NeRF-based methods and pixel-wise Gaussian feed-forward networks across all settings with 4, 8 and 12 reference views. Notably, as the number of input views increases, the reconstruction performance of both pixelSplat (Charatan et al., 2023) and MVSplat (Chen et al., 2024) degrades significantly, while AdaptiveGaussian shows a slight improvement. This is because previous methods directly merge multiple views by back-projecting pixel-wise Gaussians to 3D space based on depth maps. Without the capability to adapt the quantity and distribution of Gaussians dynamically, pixel-wise methods often produce duplicated Gaussians with significant overlap, and their spatial positioning is suboptimal. In contrast, AdaptiveGaussian is able to optimize both the distribution and quantity of Gaussians via CGA, while IGR blocks facilitate direct interaction between Gaussian queries and local image features, resulting in more accurate reconstructions.

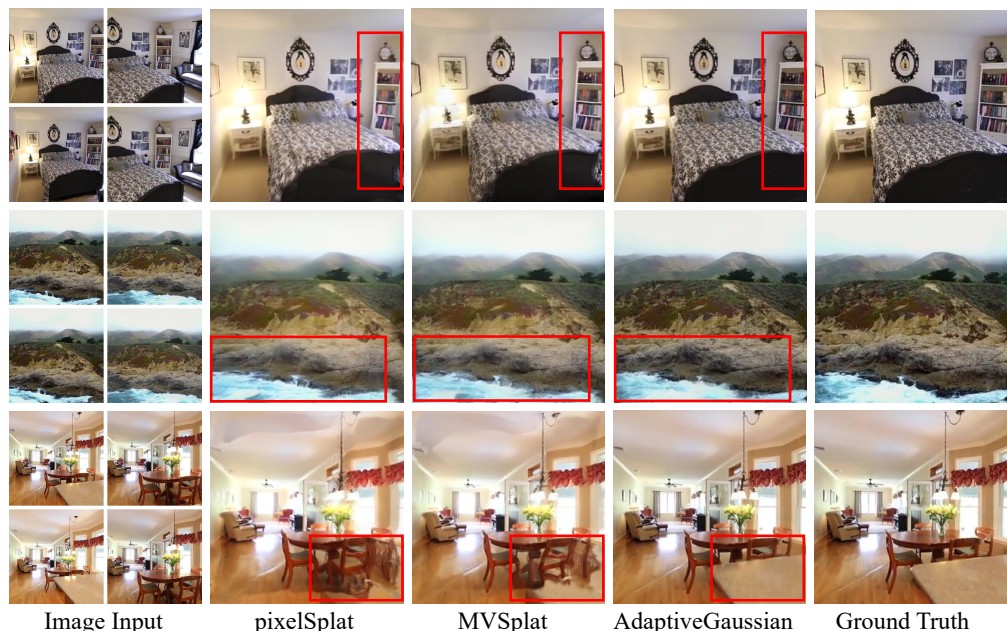

| Image Input | pixelSplat | MVSplat | AdaptiveGaussian | Ground Truth |

Figure 4: **Visualization results on ACID and RealEstate10K benchmarks.** Pixel-wise methods suffer from Gaussian overlap due to suboptimal Gaussian distributions, whereas AdaptiveGaussian enables dynamic Gaussian adaption and improved local geometry refinement.

Table 2: **Comparison of PSNR and Gaussian Quantity on RealEstate10K Dataset.** We present the average PSNR and the number of Gaussians (K) for inference using 4, 8 and 16 input views.

| Methods | 2→4 Views | | 2→8 Views | | 2→16 Views | |
|---|---|---|---|---|---|---|
| | PSNR↑ | # Gaussians | PSNR↑ | # Gaussians | PSNR↑ | # Gaussians |
| pixelSplat | 22.02 | 786 K | 19.97 | 1572 K | 18.90 | 3146 K |
| MVSplat | 22.30 | 262 K | 20.39 | 524 K | 19.40 | 1049 K |
| AdaptiveGaussian | **23.95** | **240 K** | **24.05** | **375 K** | **24.24** | **568 K** |

**Multi-View Comparison.** We further compare model performance and Gaussian quantities of different methods across various input views in Table 2. Though we find that our method requires more Gaussians than MVSplat (Chen et al., 2024) with 2 input views due to more frequent splitting than pruning, it achieves better reconstruction with fewer Gaussians as the number of views increases. In regions with richer geometric details, CGA blocks first split more Gaussians for finer representations, followed by IGR to further refine these Gaussians using deformable attention on local image features to better capture and reconstruct geometric details. Meanwhile, CGA prunes duplicate and overlapping Gaussians across views to control the growth of overall Gaussian quantity as the number of input views increases.

**Efficiency Analysis.** We explore the efficiency of AdaptiveGaussian compared with dominant pixel-wise methods on a single NVIDIA A6000 GPU. All models are inferred with multiple settings of input views on RealEstate10K (Zhou et al., 2018) dataset. We report the average inference latency and rendering FPS in Table 3. Undeniably, AdaptiveGaussian requires higher inference latency than MVSplat (Chen et al., 2024) due to the extra cost of CGA and IGR blocks. However, our model can achieve siginificantly higher rendering FPS by utilizing fewer Gaussians as the input view increases. This advantage is particularly important when rendering a large number of novel views, and it mitigates the weakness of AdaptiveGaussian on inference efficiency to some degree.

### 4.3 EXPERIMENTAL ANALYSIS

In this section, we further investigate and conduct experiments to demonstrate the effectiveness of our AdaptiveGaussian. We first visualize the both depth map and Gaussian distribution. Then, we

Table 3: **Results of Novel View Synthesis on RealEstate10K and ACID Benchmarks.** The inference time (in seconds) and rendering FPS are reported for models trained with 2 reference views and inferred with 4, 8, 12, and 16 reference views.

| Methods | 4 Views | | 8 Views | | 12 Views | | 16 Views | |
|---|---|---|---|---|---|---|---|---|
| | Inf. Time | FPS | Inf. Time | FPS | Inf. Time | FPS | Inf. Time | FPS |
| pixelSplat | 0.299 | 110 | 0.847 | 64 | 1.853 | 45 | 2.938 | 37 |
| MVSplat | 0.126 | 197 | 0.363 | 133 | 0.775 | 108 | 1.240 | 83 |
| PixelGaussian | 0.235 | 207 | 0.705 | 187 | 1.179 | 175 | 2.053 | 162 |

Figure 5: **Visualization of depth map and point cloud on multi-view NVS on RealEstate10K dataset.** AdaptiveGaussian enables to capture detailed local geometry while mitigating Gaussian redundancy across views.

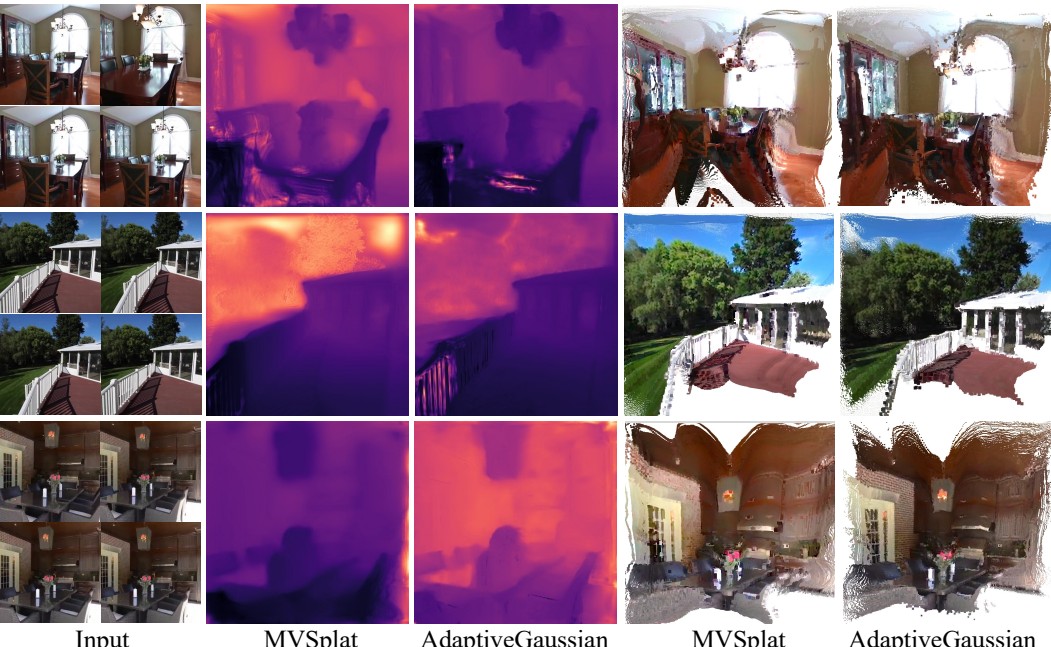

| Input | MVSplat | AdaptiveGaussian | MVSplat | AdaptiveGaussian |

conduct cross-dataset generalization and ablation studies on our model. These experiments demonstrate that CGA dynamically adapts both the distribution and quantity of Gaussians according to geometric complexity, while IGR further extract local features via direct image-Gaussian interactions, offering significant improvements over traditional pixel-wise methods.

**3D Geometry Reconstruction.** To demonstrate that our AdaptiveGaussian outperforms to reconstruct intricate local geometry as our model is able to adapt Gaussian distributions according to geometry complexity, we visualize the depth map and centers of Gaussians in Figure 5. MVSplat Chen et al. (2024) uniformly predicts pixel-aligned 3D Gaussians across images and merge the representations from different viewpoints directly, which leads to redundancy within overlapping regions and fails to fully capture the fine 3D geometry. The visual results demonstrate that the adaptive allocation and refinement processes within both the CGA and IGR blocks of our AdaptiveGaussian model generate more precise Gaussian locations, which further enhances the capability of Gaussian representations to capture the intricate 3D geometry during reconstruction.

**Cross-dataset Generalization.** To further demonstrate the generalization capability of AdaptiveGaussian, we conduct additional cross-dataset experiments. Specifically, We train our model on RealEstate10K Zhou et al. (2018) dataset and evaluate its performance on ACID Liu et al. (2021a) and DTU Jensen et al. (2014) datasets. For each setting, the reference views are sampled to ensure the coverage of the widest possible field of view. As shown in Table 4, AdaptiveGaussian is able to maintain the advantage from mitigating Gaussian overlap and redundancy in cross-dataset generalization, which leads to superior performance as input view increases.

Table 4: **Cross-dataset generalization on ACID and DTU datasets.** We sample the reference views to cover as wide a range as possible on both datasets.

| Method | ACID | | | DTU | | |
|---|---|---|---|---|---|---|
| | 4 Views | 8 Views | 16 Views | 4 Views | 8 Views | 16 Views |
| pixelSplat | 21.60 | 18.75 | 18.23 | 12.30 | 11.94 | 11.47 |
| MVSplat | 21.88 | 19.45 | 18.94 | 12.45 | 12.10 | 11.55 |
| AdaptiveGaussian | 26.01 | 26.22 | 26.37 | 13.42 | 13.46 | 13.56 |

Table 5: **Ablations on the components of AdaptiveGaussian.** We report the average PSNR, LPIPS, and the number of Gaussians (K) of model inference.

| Methods | PSNR↑ | LPIPS↓ | #Gaussians |
|---|---|---|---|
| Vanilla | 20.07 | 0.279 | 262 K |
| + Rigid Cascade Gaussian Adapter | 21.56 | 0.224 | 226 K |
| + HyperNetworks $\mathcal{H}$ | 23.07 | 0.188 | **240 K** |
| + Iterative Gaussian Refiner | **23.95** | **0.157** | **240 K** |

**Deformable Attention.** We adopt deformable attention to obtain Gaussian scores and refine Gaussian representations in both CGA and IGR blocks. To further investigate the benefits of this design, we compare the results with and without the deformable learnable keypoints generated from the query points. Since the Gaussian representations from AdaptiveGaussian are not strictly pixel-aligned, the projection of Gaussian center is uncertain to match the corresponding location in the feature maps. Deformable attention enables more flexible and adaptive Gaussian-image interactions compared to attention with rigid perception fields. Therefore, the introduction of deformable attention can lead to a PSNR increase of 1.58 in average.

**Ablation Study.** To further investigate the architecture of AdaptiveGaussian, we conduct ablation studies by inferring our model on RealEstate10K (Zhou et al., 2018) test dataset with 4 input views. We first introduce a vanilla model, where the initial Gaussian set $\mathcal{G}$ is directly used to render novel views. Then, we adopt rigid CGA blocks without context-aware Hypernetworks $\mathcal{H}$, which means Gaussian set $\mathcal{G}$ goes through splitting and pruning based on fixed thresholds ($\tau_{high}^{(k)} = 0.8, \tau_{low}^{(k)} = 0.2, k = 1, 2, ..., K$). We further add HyperNetworks $\mathcal{H}$ to generate context-aware thresholds. Finally, we adopt IGR blocks to refine the Gaussian representations via image-Gaussian interactions. As shown in Table 5, HyperNetworks $\mathcal{H}$ utilizes score maps $\mathcal{S}$ to generate context-aware thresholds, enabling a more dynamic and efficient Gaussian distribution for scene representation compared to rigid splitting and pruning. Furthermore, IGR blocks refine the Gaussian set iteratively via deformable attention between Gaussians and image features, enhancing their ability to describe and reconstruct intricate local geometric details.

## 5 CONCLUSION

In this paper, we have presented AdaptiveGaussian to learn generalizable 3D Gaussian reconstruction from arbitrary input views. AdaptiveGaussian is able to dynamically adapt both Gaussian distribution and quantity guided by the complexity of local geometry details in the Cascade Gaussian Adapter blocks, which allocate more to detailed regions and reducing redundancy across views. Further, Iterative Gaussian Refiner blocks facilitate direct image-Gaussian interactions to improve local geometry reconstructions. thus leading to superior performance in reconstruction and view synthesis compared to pixel-wise paradigm.

**Discussions and Limitations.** Although AdaptiveGaussian can adjust the distribution of 3D Gaussians dynamically, the initial Gaussians are still derived from pixel-wise unprojection. When we initialize the Gaussian centers completely at random, the model fails to converge. Moreover, deformable attention in IGR consumes substantial computational resources when the number of Gaussians is extremely large, highlighting the need for a more efficient approach to represent 3D scenes with fewer Gaussians. Furthermore, AdaptiveGaussian is unable to perceive the unseen parts of 3D scenes beyond the input views, suggesting the potential need to incorporate generative models.

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
