# OpenReview forum: "PixelGaussian: Generalizable 3D Gaussian Reconstruction from Arbitrary Views"
_ICLR.cc/2025/Conference — Submitted to ICLR 2025_

### Official Review · Reviewer_X5K2 · 2024-10-25

**Soundness:** 3
**Presentation:** 3
**Contribution:** 3
**Rating:** 6
**Confidence:** 5

**Summary:**

This work first points out the unique issue posed by the use of pixel-aligned 3D Gaussians, which is the redundant 3D Gaussians depicting the same obejcts' surfaces. Due to this issue, existing works, such as MVSplat and PixelSplat, perform bad when the number of views increases, unlike the NeRF-based generalized NVS works, PixelNeRF, IBRNet or MVSNeRF, that benefit from more input views. To handle this issue, the authors propose so called CGA, to adjust Gaussian distribution according to local geometry complexity identified by a keypoint scoring function. Subsequently, a transformer-based IGR is proposed to model interactions between image and 3D Gaussians to realize further refinements. The proposed method is evaluated on large-scale indoor/outdoor datasets and is shown to achieve new state-of-the-art.

**Strengths:**

1. Motivations are clear. PixelSplat and MVSplat indeed have problems when the number of views increase. This paper tackles this issue, which I think is very reasonable.

2. Ablation study is thorough and the performance of the proposed method is impressive.

3. Writing is good and easy to understand.

**Weaknesses:**

1. It seems that the performance of PixelGaussian is better in 2 view setting as well. Why is this? I understand that for more views, it will the proposed method will perform better than them, but i can not think of a convincing reason for the superior performance in 2 view setting. I am guessing that the model is trained on 8 A6000 GPUs, which incorporates larger effective batch size than MVSplat. So I would say it's not fair to compare directly with MVSplat, thus we can not be so sure that the proposed method would work also well with 2 views. It might even perform worse if trained with similar iterations and batch size. Would it be possible if the authors  provide a more detailed comparison of training conditions between PixelGaussian and MVSplat, including batch sizes, number of iterations, and computational resources used. Additionally, if PixelGaussian trained under similar conditions as MVSplat to ensure a fair comparison in the 2-view setting, what would be the results?

2. In Table 3, the authors highlight that the rendering FPS is higher than MVSPlat. This makes sense, since the # of Gaussians are less compared to MVSplat. However, Since the architecture of the proposed method builds on top of MVSplat, it will likely to infer slower overall, if both 3D reconstruction and nvs were accounted for, which is reported and supported by the latency. So I think the efficiency is not really a big contribution, **unless** the authors showed the differences in higher resolutions. Currently 256x 256 is too small to see the dramatic difference. But the current submission unfortunately does not have efficiency comparisons in higher resolutions, which could've been a strength of this work.  Would it be possible if the authors provide detailed comparisons at higher resolutions (e.g., 512x512, 1024x1024 or higher).  This could be this paper's strength if properly compared.

3. It would've been even more interesting if the paper explored the impacts of increasing the number of input views more and more. I would expect at some point, the performance will saturate, but it will be very interesting to see those. Would it be possible if the authors systematically increases the number of input views (e.g., from 2 to 8 or 10) and report how the performance changes, including where it potentially saturates?

**Questions:**

Please see the above weaknesses. If they are adequately addressed, I am willing to increase the rating.

---

> ### Author Response · Authors · 2024-11-25
>
> Thank you for your valuable comments. Below we address specific questions.
>
> **1. Training Setting and Result on 2 Views**
>
> Thank you for your valuable question. You are correct that there are differences in the training configurations between pixelSplat, MVSplat, and PixelGaussian in the current results. Therefore, we retrain pixelSplat, MVSplat and PixelGaussian on RealEstate10K dataset using the same setting to ensure a fair comparison. The results of common cases and challenging cases on RealEstate10K dataset are listed in the Table 1 below and in our the general rebuttal, respectively. Additionally, we offer the detailed training cofiguration for all the models in Table 2 for your reference.
>
> Table 1: PSNR comparison of multiple input view settings on RealEstate10K validation dataset
> |          | pixelSplat | MVSplat | PixelGaussian |
> |----------|----------- |---------|--------------|
> | 2 views | 25.07 | 25.80 | 25.68 |
> | 4 views | 22.26 | 22.77 | 25.70 |
> | 8 views | 20.01 | 20.62 | 25.78 |
> | 16 views | 19.47 | 19.88 | 25.83 |
>
> Table 2: Detailed training configuration of pixelSplat, MVSplat and PixelGaussian
> | Configuration | Setting |
> |---------------|---------|
> | GPUs | 1 NVIDIA A6000 |
> | batch size | 4 |
> | iteration | 300,000 |
> | optimizer | Adam |
> | scheduler | Cosine Annealing |
> | learning rate | 1e-4 |
> | weight decay | 5e-5 |
>
> From the results, we do observe that MVSplat slightly outperforms PixelGaussian in the 2-view input setting. We have corrected and clarified this in our revised version. However, as the number of input views increases, PixelGaussian still demonstrates a significant advantage due to the Gaussian overlap and redundancy can be effectively mitigated through our adaptive refinement process. We appreciate it that you point out this issue.
>
> **2. Inference Efficiency**
>
> Thank you for your comment. As illustrated in Table 3 in our origin paper, our PixelGaussian requires higher inference latency and memory due to the extra CGA and IGR blocks, while can achieve a higher rendering FPS since the quantity of Gaussians is smaller. We further conduct experiments to analysis the training efficiency, inference efficiency and rendering FPS of our network with multiple input view and resolution settings. We use a 48G A6000 and set the batch size as 1 uniformly.
>
> Table 3: Training FPS
> | pixelSplat | MVSplat | PixelGaussian |
> |-------|-------|-------|
> | 2.26 | 8.20 | 3.52 |
>
> Table 4: Inference atency (ms) on multiple view settings
> |         | pixelSplat | MVSplat | PixelGaussian |
> |---------|------------|---------|---------------|
> | 2 views | 136.6 | 60.8  | 107.3 |
> | 4 views | 299.4 | 126.4 | 235.0 |
> | 6 views | 563.8 | 237.3 | 415.8 |
> | 8 views | 846.5 | 363.2 | 704.6 |
>
> Table 5: Inference latency (ms) and on multiple resolution settings with 2 input views
> |           | MVSplat | PixelGaussian |
> |-----------|---------|---------------|
> | 256x256   | 60.8   | 107.3  |
> | 512x512   | 235.0  | 432.5  |
> | 1024x1024 | 943.3  | 1725.0 |
>
> Table 6: Rendering FPS on multiple view settings
> |         | MVSplat | PixelGaussian |
> |---------|------------|---------|
> | 4 views | 197 | 207 |
> | 8 views | 133 | 187 |
> | 12 views | 108 | 175 |
> | 16 views | 83 | 162 |
>
> Table 7: Rendering FPS and on multiple resolution settings with 4 input views
> |           | MVSplat | PixelGaussian |
> |-----------|---------|---------------|
> | 256x256   | 197 | 207 |
> | 512x512   | 84 | 142 |
>
> As illustrated in Table 4 and 5, our proposed PixelGaussian does require higher overall latency for training and inference compared to MVSplat due to the extra blocks and we also acknowledge that efficiency is not a primary contribution of this paper when considering both 3D reconstruction and NVS. However, Tables 6 and 7 demonstrate the significant advantage of PixelGaussian on rendering at higher resolutions and with increasing input views. This is particularly important when rendering a large number of novel views, and it even helps mitigate the weakness of inference efficiency. Additionally, the adaptation and refinement in PixelGaussian can effectively address the issues of Gaussian overlap and redundancy in both pixelSplat and MVSplat. Thank you for your insightful suggestion and we have added this explanation in our revised paper.
>
> **3. Model Performance as Input View Increases**
>
> The performance saturation you mentioned is quite insightful. In our general rebuttal, we have conducted experiments with additional view settings across multiple datasets. Due to the memory limitation, we are only able to inference our model with a maximum of 16 views as input. From the results, we do observe that as the number of input views increases, the performance gains gradually diminish, which suggests the presence of performance saturation. Therefore, it is crucial to find a balance between the increasing computational overhead and the marginal performance improvements in practical applications.

---

> > ### Comment · Reviewer_X5K2 · 2024-11-27
> >
> > Thank you for the detailed response.
> >
> > All of my concerns/questions are adequately addressed.
> >
> > I will raise the rating to 6.

---

### Official Review · Reviewer_z7ms · 2024-10-28

**Soundness:** 3
**Presentation:** 3
**Contribution:** 3
**Rating:** 5
**Confidence:** 5

**Summary:**

PixelGaussian employs a dynamic mechanism to identify the region of interest within the underlying 3D spatial locations, and accordingly adjusts the 3D-GS distribution and quantity to enhance visual quality while reducing memory usage. Experiments on ACID and RealEstate10K validate the effectiveness of the proposed method.

**Strengths:**

1. The proposed CGA and IGR modules are novel in the feed-forward 3D-GS scenarios.
2. The experiments effectively support the proposed method: Table 1 demonstrates visual performance compared to SOTAs, Table 2 illustrates scalability concerning input views, Table 3 highlights inference efficiency achieved through reduced 3D Gaussians, and Table 4 validates the effectiveness of the designed modules.
3. The paper is well-motivated and easy to understand and follow.

**Weaknesses:**

Please see the "Questions" section.

**Questions:**

1. As shown in Table 1, the 3D-GS-based PixelGaussian demonstrates marginally improved performance over the NeRF-based MuRF, especially on the ACID benchmark. A more detailed analysis is required to explain the underlying reasons for this performance difference, training and inference efficiency, and memory usage.
2. RealEstate10K and ACID benchmarks were first officially introduced in pixelSplat, but I think the efficiency should be demonstrated in more benchmarks. I would like to see more comparisons in more diverse datasets, such as MuRF's DTU, LLFF, and Mip-NeRF 360 dataset, or even self-captured real-world via cameras or phones (see the teaser in Metric3D [1] and F2-nerf [2]).
3. I would like to see the full efficiency comparison across 2-6 views, in addition to only 4 views in Table 3.
4. The training efficiency comparison with MVSplat should also be analyzed with the same amount of training data and computing resources.
5. I think the 3D-GS adaptions in Fig. 6 and Fig. 1 are not visually-pleasing in the ICLR-level paper and can be drawn and demonstrated better.


[1] Metric3d: Towards zero-shot metric 3d prediction from a single image. ICCV 2023

[2] F2-nerf: Fast neural radiance field training with free camera trajectories. CVPR 2023

---

> ### Author Response · Authors · 2024-11-25
>
> Thank you for your insightful comments. Below we address specific questions.
>
> **Comparison with NeRF-based Models**
>
> Thank you for your comment. We provide a more details comparsions between NeRF-based methods and our PixelGaussian in the following table. All experiments are conducted on the challenging cases on RealEstate10K as described in our general rebuttal with the batch size of 1 on a single A6000 GPU.
>
> Table 1: Detailed comparisons on PSNR and inference time with NeRF-based methods.
>
> | Views     | Method         | PSNR  | Inference Time (s)  |
> |-----------|----------------|-------|---------------------|
> | 4 views   | pixelNeRF      | 21.03 | 5.30 |
> |           | MuRF           | 23.30 | 0.36 |
> |           | PixelGaussian  | 23.95 | 0.24 |
> | 8 views   | pixelNeRF      | 21.22 | 5.77 |
> |           | MuRF           | 23.78 | 0.75 |
> |           | PixelGaussian  | 24.05 | 0.70 |
> | 12 views  | pixelNeRF      | 21.28 | 6.04 |
> |           | MuRF           | 23.94 | 1.34 |
> |           | PixelGaussian  | 24.18 | 1.19 |
> | 16 views  | pixelNeRF      | 21.30 | 6.48 |
> |           | MuRF           | 24.16 | 2.09 |
> |           | PixelGaussian  | 24.24 | 1.84 |
>
> As shown in the results, NeRF-based methods do not experience the Gaussian overlap and redundancy issues as it is in pixelSplat and MVSplat, which explains why they can slightly benefit from an increase in input views similar to PixelGaussian. However, thanks to the efficient rasterization-based rendering of Gaussians, PixelGaussian demonstrates superior computational efficiency compared to NeRF-based methods.
>
> **More comparisons on Multiple Datasets and Settings**
>
> Thanks for the suggestions. We have conducted experiments on challenging cases with multiple view settings across a wider range of scenarios in our general rebuttal, which have demonstrated that our PixelGaussian can consistently benefit from the increase of input views across all experimental settings. We have included these results in our revised paper.
>
> **Training and Inference Efficiency**
>
> We agree with you that additional comparisons on training and inference efficiency are essential. We further conduct experiments to analysis the training and inference efficiency of our network with multiple input view settings. We use a 48G A6000 and set the batch size as 1 uniformly.
>
> Table 1: Training FPS
> | pixelSplat | MVSplat | PixelGaussian |
> |-------|-------|-------|
> | 2.26 | 8.20 | 3.52 |
>
> Table 2: Inference Latency (ms) on multiple view settings
> |         | pixelSplat | MVSplat | PixelGaussian |
> |---------|------------|---------|---------------|
> | 2 views | 136.6 | 60.8  | 107.3 |
> | 4 views | 299.4 | 126.4 | 235.0 |
> | 6 views | 563.8 | 237.3 | 415.8 |
> | 8 views | 846.5 | 363.2 | 704.6 |
>
> Table 3: Rendering FPS on multiple view settings
> |         | MVSplat | PixelGaussian |
> |---------|------------|---------|
> | 4 views | 197 | 207 |
> | 8 views | 133 | 187 |
> | 12 views | 108 | 175 |
> | 16 views | 83 | 162 |
>
> Our proposed PixelGaussian does require higher overall latency for training and inference compared to MVSplat due to the extra blocks. However, inspired by reviewer X5K2, we have found that PixelGaussian achieves significantly higher rendering speeds as the number and resolution of input views increase. This becomes particularly valuable when rendering a large number of novel views, and it can make up to the limitations of our inference speed.
>
> **Figure**
>
> Thank you for your advice. We have revised the figure you mentioned and added both depth and point cloud visualization to our revised paper.

---

> ### Comment · Reviewer_z7ms · 2024-12-01
>
> Thanks for the authors' response, however
>
> 1. It seems my question regarding W1 hasn’t been fully addressed, as the comparison of training efficiency and memory usage has not been presented quantitatively. I would like a more comprehensive comparison, as the performance improvement is not particularly evident.
>
> 2. Regarding the additional experiments on DTU, why do all the methods perform poorly? I believe that a visual quality below 15 PSNR indicates significant issues. This suggests that the method does not generalize well to different data settings. Additionally, why was evaluation not performed on MipNeRF-360? I think this wouldn’t require extra training, just an adjustment of the camera intrinsics/extrinsic if the aspect ratio is incorrect. Was the performance of this dataset not satisfactory?
>
> 3. Regarding the comparison of Training FPS/Inference Latency, I understand that the additional CGA and IGR modules introduce extra overhead. However, I’m still disappointed that PixelGaussian performs similarly or perhaps even worse than MVSplat in rendering FPS when using fewer than 4 views. As we know, sparse views are typically analyzed with 5 or fewer views, so the performance difference in denser-view settings should be considered a bonus rather than a main focus. Moreover, the fact that PixelGaussian lags behind MVSplat in latency speed is concerning.
>
> 4. Figure 5 still doesn’t convincingly demonstrate the performance gain and is not beautiful in terms of ICLR-level presentation.

---

> ### Comment · Reviewer_z7ms · 2024-12-01
> **Final Decision**
>
> I have reviewed all the comments and replies from the other reviewers. For me, the main issue is as follows:
>
> The authors propose the CGA and IGR modules to reduce 3D-GS redundancy, but these additions introduce extra overhead, leading to higher inference latency compared to MVSplat. Furthermore, in the test setting with fewer than 4 views, the rendering FPS of PixelGaussian is also similar or perhaps even worse than MVSplat, which is a more critical scenario than dense-view NVS. These points suggest that the design of the CGA and IGR modules is not sufficiently effective and requires further elaboration.
>
> Considering these concerns and the rebuttal's inability to fully address them, I am finalizing my score at **5, marginally below the acceptance threshold, with confidence 5**.
>
> (Extra message)
> 1. The term 'PixelGaussian' and 'AdaptiveGaussian' mix together (e.g., Table 3 and Table 4), which is a cheap mistake.
> 2. Do not leave the reply on the last minutes/days even if you are occupied with CVPR submission in that period.

---

> ### Comment · Reviewer_X5K2 · 2024-12-01
>
> Dear Reviewer z7ms,
>
> Thank you for the detailed reviews and comments. While I agree with some of your points, I'm wondering why would it be so critical not to introduce additional latency compared to MVSplat? As far as I understood, the inference speed should not be of much importance in this work, given that under 4 view settings, it is not significantly slower than MVSplat, and the main focus is on scenarios when more views are provided. Also, DTU is known for its artificial setup and far from real world scenarios, which makes it hard to conclude that the method does not generalize well. It would be better to decide whether the method generalized well or not by looking at datasets like DL3DV.
>
> The performance gain when N>5 shouldn't be a bonus since it is shown by PixelGaussian that other methods clearly struggle with more views.

---

> ### Comment · Reviewer_z7ms · 2024-12-01
>
> From my perspective,
>
> (1) The generalizable 3D-GS setting is more plausible for few views (e.g. 2 views in PixelSplat/MVSplat). I'm not fully convinced by the point of "main focus is on scenarios when more views are provided" in the feed-forward setting. If you are feeding in more views as input, why not directly compare with per-scene optimized 3D-GS methods? These methods could likely achieve better PSNR, albeit at the cost of longer training times. The paper primarily proposes the CGA and IGR modules to reduce GS redundancy, but they do not provide sufficient evidence of a performance gain ("not significantly slower than MVSplat") in rendering FPS for the **crucial settings with fewer views (<4, such as 2 or 3)**.
>
> (2) I acknowledge that "DTU is known for its artificial setup", so why not consider adopting my suggestion to evaluate more real-world-like datasets, such as the "Mip-NeRF 360 dataset," or "even using self-captured real-world inputs"? I suggest the Mip-NeRF 360 dataset because I’d like to see how the model performs on less forward-facing inputs when generalizing. As for the self-captured real-world inputs, I’m interested in seeing how the model applies to real-world scenarios (please refer to the DepthSplat project page for some examples).
>
> (3) I understand that rendering FPS is much more important than "inference latency", but since you put the latency comparison in the main paper, it naturally raises my concern regarding the effectiveness of architecture design. For rendering FPS, please see (1).

---

> ### Comment · Reviewer_X5K2 · 2024-12-01
>
> Thank you for the quick reply!
> I agree with your points 2 and 3. However, just one last comment I would like to make.
> "These methods could likely achieve better PSNR, albeit at the cost of longer training times", the cost of longer training times is a significant disadvantage in many scenarios and I think it would not be ideal to suggest to compare with per-scene optimization approaches. N<12 views are still considered sparse views and typically, the 2 view settings of Pixelsplat or MVSplat would be considered less practical, as 2 view settings is the most basic multi view geometry setting.
>
> I believe that since the paper attempts to address the apparent limitations of existing pixel-aligned Gaussians (redundancy), we should focus on this aspect rather than putting much focus on  the performance directly compared to per scene optimizations or SOTA.
>
> Nevertheless, I agree with your other points, which I appreciate your time for the discussion with me.

---

> ### Comment · Reviewer_z7ms · 2024-12-01
>
> (1) I’m not demanding that you compare with per-scene optimized 3D-GS methods; I simply use this as an illustration to show that emphasizing more dense-view input in feed-forward 3D-GS is not inappropriate.
>
> (2) I still believe that '12-view' is too large for the input in a sparse-view setting. While I’m confident that dense-view inputs can support the generalizable properties of the proposed method, they should not be the main selling point, as sparse-view inputs are more critical in this context.
>
> (3) I disagree with the idea that “2-view settings are less practical”. In fact, exploring a one-shot setting could be both more interesting and more challenging. Practically speaking, if you were to take pictures with your phone for 3D scene reconstruction, it would be more convenient to take just 1-3 shots from different viewpoints. Dynamic factors like people moving or changing environmental lighting could introduce interference, and feeding in additional views might worsen inconsistencies.
>
> Since you have your own perspective on this, I’ll leave it to the AC to make the final judgment. Thanks for the discussion.

---

> > ### Author Response · Authors · 2024-12-03
> >
> > **1.Training Efficiency**
> >
> > Thank you for your comment. We have reported the training efficiency comparisons in our initial reply. Here, we provide a summary of both training efficiency and memory usage on a A6000 GPU with the batch size of 1 for your reference. As indicated in our paper, AdaptiveGaussian does require more training memory and latency than MVSplat, however, it is still demonstrated to be more efficient than pixelSplat.
> >
> > Table 1: Training FPS and memory usage (GB)
> > |       | pixelSplat | MVSplat | PixelGaussian |
> > |-------|-------|-------|-------|
> > | FPS | 2.26 | 8.20 | 3.52 |
> > | Memory | 18.82 | 5.86 | 7.85 |
> >
> > **2.MipNeRF-360**
> >
> > The poor performance on DTU is primarily due to the significant gap compared to the real-captured ACID and RealEstate10K training datasets. Following your suggestion, we evaluated on the MipNeRF-360 dataset. The results shown below still demonstrate that AdaptiveGaussian continues to benefit from an increased number of input views. In contrast, both pixelSplat and MVSplat experience Gaussian redundancy, leading to a dramatic performance drop.
> >
> > Table 2: PSNR comparison of multiple input view settings for cross-dataset generalization on MipNeRF-360 dataset
> > |          | pixelSplat | MVSplat | AdaptiveGaussian |
> > |----------|------------|---------|---------------|
> > | 4 views | 21.60 | 21.84 | 23.47 |
> > | 8 views | 19.39 | 19.70 | 23.56 |
> > | 12 views | 17.27 | 17.38 | 23.61 |
> >
> > **3.View Count**
> >
> > We appreciate the feedback and discussions from all reviewers and would like to clarify our stance.
> >
> > We **did not** claim that AdaptiveGaussian is only suitable for sparse view reconstruction anywhere in our paper. Our main contribution is to enable **feed-forward** 3D Gaussian reconstruction from an **arbitrary number of input views**, while our model can also benefit from an increased number of input views. Although, compared to the original setting of 3D Gaussian Splatting, which requires hundreds of input views to optimize the final scene, the number of views in our experimental setup is relatively sparse. However, this limitation is solely due to the memory constraints, not to mention the efficiency and generalizability advantages that feed-forward frameworks offer over optimization-based methods.

---

### Official Review · Reviewer_HfoT · 2024-10-28

**Soundness:** 3
**Presentation:** 2
**Contribution:** 3
**Rating:** 6
**Confidence:** 4

**Summary:**

They propose a feed-forward framework 3D Gaussian reconstruction from arbitrary views. With propose CGA and IGR, they can produce well distributed gaussians base on geomtry complexity and generalize better to arbitrary views than other methods. They have validation on benchmarks to show the effectiveness of their method.

**Strengths:**

1.The paper proposes a generalizable gaussian prediction method with Cascade Gaussian Adapter (CGA) to dynamically adapt the distribution and quantity of Gaussians, which is effectively shown in the experiments part.

2.It further refines Gaussian representations via proposed Iterative Gaussian Refiner with deformable attention.

3.It shows with better distributed gaussians, the final rendering results are better than those methods with per-pixel predicted gaussians and better generalization when trained with 2 views and tested with 2-4 views.

**Weaknesses:**

1. While the paper demonstrates improved generalization with increased view numbers during inference through adjusted Gaussian distributions. However, it would make more sense by including results for baseline methods trained specifically on 2, 4, and 6 views.

2. The current experiments primarily showcase scenarios with substantial FOV overlap between input views. To better demonstrate the method's generalization capabilities, consider including more challenging test cases - for instance, evaluating a model trained on 2 views when tested with 2-6 widely-spaced views with minimal overlap. Such examples would validate the method's ability to handle challenging novel view synthesis scenarios.

**Questions:**

1.Is it possible to provide details on the view selection process, including the angular separation between views and whether test views are interpolated between or extrapolated beyond the training views as well as the camera pose distribution for both training and testing sets.

2.Since the author argues their method can be trained on two views and tested on more views. Is it possible to show that such training strategies have better generalization ability for large scale/wide range scenes and compare results with those diffusion-based sparse view reconstruction methods including ReconFusion, ViewCrafter, ReconX following their experiments settings. Also give more discussions on such settings.

---

> ### Author Response · Authors · 2024-11-25
>
> # Reviewer HfoT
>
> Thank you for your valuable comments. Below we address specific questions.
>
> **1. View Setting for Training**
>
> We agree with you that training baseline models on different view settings could potentially lead to better results. As you suggested, we are still training the model using 4 and 8 views as input. However, we argue that it is more user-friendly to use a single model regardless the number of input views. This is why our goal is to achieve generalizable reconstruction from **arbitrary views**.
>
> **2. Results on challenging scenarios**
>
> We agree that most scenes have substantial FOV overlap. To demonstrate the performance of our model on NVS across a wider range of scenes, we conduct experiments on challenging cases in both the RealEstate10K and DTU datasets in our general rebuttal. As you suggested, we have included these experiments in our revised paper.
>
> **3. View Sampling**
>
> All the novel views are sampled and interpolated within the range of reference views. As the number of input view increases, the reference views cover a wider range of FoV. For the hard cases illustrated in the general rebuttal, we sample the reference views to cover as wide a view range as possible.
>
> **4. Generalization on Large-scale and Wide-range scenes**
>
> Your suggestion is quite insightful. As we point out in our Discussion section, the introduction of generative model can address unseen areas of the reference views, enabling NVS for extrapolated perspectives. Additionally, diffusion-based methods can leverage the strong diffusion priors and achieve a boost in NVS for interpolated views (such as ReConX can achieve a PSNR of 28.31 on the RealEstate10K validation set). Since we are not able to access the code of ReConX and ReconFusion, we only compare our PixelGaussian with the NVS results from the video diffusion model in ViewCrafter on 5 selected scenes in the RealEstate10K validation set.
>
> |       | ViewCrafter | PixelGaussian |
> |-------|-------|-------|
> | interpolated | 24.31 | 27.47 |
> | extrapolated | 22.90 | 19.75 |
>
> We see that with the strong diffusion priors, ViewCrafter achieves better performance on extrapolated view settings. Your valuable insight inspires us to combine our adaptive Gaussian pipeline with video diffusion models, which may potentially lead to remarkable results on reconstruction and NVS for large-scale and wide-range scenes.

---

> > ### Comment · Reviewer_HfoT · 2024-11-26
> > **Reply to authors**
> >
> > 1."View Setting for Training": I don't see a comparison between your method and pixelSplat/MVSplat when trained with 4,6,8 views. Could you please point out where this comparison is shown in your results? While user-friendly view input is beneficial, it cannot be considered a significant contribution if the model's performance degrades compared to existing methods when trained on the same number of views. Additionally, to demonstrate superiority, it would be valuable to see comparisons where all models (including yours and competing methods) are trained on 8 views but tested on 2, 4, and 6 views.
> >
> > 2. "Results on challenging scenarios": can you please specify which figure is the results of "model trained on 2 views when tested with 2-6 widely-spaced views with minimal overlap" as I suggested? Also specify the camera configurations instead of just give one interplation NVS. Better show in the supplementary videos compared with others for us to validate.
> >
> > 3. "View Sampling": I don't see the specific explaination of Question1.
> >
> > 4. "Generalization on Large-scale and Wide-range scenes": it is good to see the comparison with video diffusion-base method. However, it seems there is huge performance drop (27.47 to 19.75) from interpolation to extrapolation.  While this degradation is understandable given the lack of generative priors in your model, it highlights the need for comprehensive evaluation protocols. Specifically: The main experiments should include both interpolation and extrapolation results for all compared methods (pixelSplat/MVSplat) and specify how the FOV overlap is, not just interpolation. This is also the reason why I need the specified the camera configurations).
> >
> > These comprehensive comparisons would provide a more complete understanding of each method's strengths and limitations across different view synthesis scenarios. Given the insufficient expanation and experiments, I would lower the score. However, I may raise my socre if my concerns are resolved.

---

> > > ### Author Response · Authors · 2024-12-01
> > >
> > > **1. View Setting for Training**
> > >
> > > Thank you for your comment. As you suggested, we have finished the training of pixelSplat, MVSplat and our AdaptiveGaussian with 8 views as input with the same training configuration as in our main results. However, we use 8 GPUs to speed up the training process. All the models are evaluated on the 4-view, 8-view and 12-view validation set, respectively. Furthermore, we report the training time of MVSplat with multiple view settings on a single A6000 GPU with the batch size of 1.
> > >
> > > Table 1: PSNR comparison of multiple input view settings and models trained on different views on RealEstate10K
> > >
> > > |          | pixelSplat-2view | MVSplat-2view | AdaptiveGaussian-2view | pixelSplat-8view | MVSplat-8view | AdaptiveGaussian-8view |
> > > |----------|-----------|---------|--------------|-----------|---------|--------------|
> > > | 4 views | 22.02 | 22.30 | 23.95 | 22.64 | 22.98 | 23.87 |
> > > | 8 views | 19.97 | 20.39 | 24.05 | 23.59 | 23.74 | 24.09 |
> > > | 12 views | 18.92 | 19.69 | 24.18 | 21.05 | 21.47 | 24.23 |
> > >
> > > Table 2: Training time (h) of MVSplat across multiple view settings
> > > | 2view | 4view | 8view | 16view |
> > > |----------|-----------|---------|--------------|
> > > | 78 | 170 | 445 | 1475 |
> > >
> > > Although the performance of pixelSplat and MVSplat on multiple view settings improves after training on more views, our AdaptiveGaussian-2view still outperforms both frameworks. This result supports our claim that Gaussian redundancy and overlap across views can negatively impact the reconstruction quality. Additionally, AdaptiveGaussian offers a more training-efficient and view-robust approach for 3D Gaussian reconstruction from arbitrary views, while training models with a specific number of views incurs intolerable computational overhead and time (as shown in Table 2).
> > >
> > > **2. Results on challenging scenarios**
> > >
> > > Thank you for pointing out. The additional results on DTU dataset can be considered cases "with minimal overlap". We prove this by projecting the pixel-wise Gaussian centers from one reference view onto the nearest reference view following our view sampling strategy, and find only averagely **21.4%** of the points can be projected onto the latter view on the 4 view setting. For your reference, we list the evaluation result on DTU in our general rebuttal below, and we will include the visualization results for such cases if our paper is accepted.
> > >
> > > Table 3: PSNR comparison of multiple input view settings for cross-dataset generalization on DTU dataset
> > > |          | pixelSplat | MVSplat | PixelGaussian |
> > > |----------|------------|---------|---------------|
> > > | 4 views | 12.30 | 12.45 | 13.42 |
> > > | 8 views | 11.94 | 12.10 | 13.46 |
> > > | 12 views | 11.75 | 11.78 | 13.53 |
> > > | 16 views | 11.47 | 11.55 | 13.56 |
> > >
> > > **3. View Sampling**
> > >
> > > We further elaborate the view sampling process as you mentioned. For each scene in both training and test sets, reference views and inference views for NVS are sampled from a temporal video. Each video comprises approximately 270 frames on average, capturing scenes with either zoom-in, zoom-out, or surround patterns. For view sampling, we evenly select images as reference views and ensure that the reference views cover the entire range of the video. Then, we collect interpolated views as inference views for model evaluation. We evaluate the FOV overlap by projecting per-pixel predicted points from the first view onto the second view in each pair of adjacent reference frames. On average, we observe that approximately **two-thirds** of the points can be projected onto the latter view.
> > >
> > > **4. Generalization on Large-scale and Wide-range scenes**
> > >
> > > Thank you for your comment. The issue of FOV overlap and camera configuration has already been explained. As you suggested, we also evaluate the model performance on extrapolation view settings, where we ensure **a minimum two-thirds FOV overlap** between inference and reference views. All models are evaluated using 4 input views, and we include both the results on extrapolation with and without the overlap mask (i.e. masking black pixels in the inference results when calculating PSNR).
> > >
> > > Table 4: interpolation and extrapolation PSNR results comparsion on RealEstate10k
> > > |          | pixelSplat | MVSplat | AdaptiveGaussian |
> > > |----------|-----------|---------|--------------|
> > > | interpolation, w.o. mask | 22.02 | 22.30 | 23.95 |
> > > | extrapolation, w.o. mask | 17.87 | 18.40 | 20.18 |
> > > | extrapolation, mask | 21.65 | 22.07 | 23.43 |
> > >
> > > In both interpolation and extrapolation settings, AdaptiveGaussian outperforms both pixelSplat and MVSplat using 4 input views. Additionally, the NVS performance in the extrapolation setting is comparable to that in the interpolation setting when the overlap mask is applied during evaluation. Despite the significant performance drop in extrapolation compared to methods with diffusion priors, our method still outperforms those without the integration of generative models across all settings.

---

> > > > ### Comment · Reviewer_HfoT · 2024-12-03
> > > > **Reply to the authors**
> > > >
> > > > Thanks for your detailed explanations and experiments which make the proposed method more convincing. I would like to raise my score to 6.

---

### Official Review · Reviewer_bYDc · 2024-11-04

**Soundness:** 3
**Presentation:** 3
**Contribution:** 3
**Rating:** 5
**Confidence:** 5

**Summary:**

This work introduces PixelGaussian, a feed-forward 3DGS model that can adaptively update the number of Gaussian via pruning and splitting. In particular, PixelGaussian starts by predicting the pixel-aligned 3D Gaussians. These 3D Gaussians are used as input to the introduced Cascade Gaussian Adapter (CGA) and Iterative Gaussian Refiner (IGR) for further processing, yielding a set of updated 3D Gaussians aligned with the geometry complexity. Experiments on RealEstate10K and ACID demonstrate the effectiveness of PixelGaussian, showing state-of-the-art performance.

**Strengths:**

* The motivation is reasonable, emphasising the significance of updating the predicted pixel-aligned 3D Gaussians under more input views.
* Experiments on two existing benchmarks verify the efficacy of the introduced PixelGaussian

**Weaknesses:**

* Accessing the quality of depth map and extracted point cloud. As mentioned in L91, this work aims to “mitigate Gaussian overlap and redundancy”. However, this contribution cannot be justified by the RGB image-based results. It is crucial to access the 3D Gaussian quality by visualising the depth map and point cloud, similar to Fig. 4 in MVSplat. In particular, it would be better to compare with pixelSplat and MVSplat in terms of point cloud and depth map, especially using more input views, e.g., 4 input views. It is also suggested that similar comparisons be made among different ablated models, which would make it easier to understand the effectiveness of CGA and IGR.

* Accessing more input views. The significance of refining 3D Gaussians would be more obvious under the settings of more input views. Hence, it would be interesting to see how well the introduced PixelGaussian can perform using more input views, e.g., 12 views, 16 views, or even 32 views, similar to the concurrent work Long-LRM [Ziwen et. al, arXiv:2410.12781].

* Accessing more complex datasets. As shown in Fig. 4, the visual differences between PixelGaussian and other state-of-the-art methods are minor, possibly because RealEstate10K is too simple to demonstrate the effectiveness of Gaussian refinement. Hence, it is highly recommended to report additional comparisons on more complex datasets, e.g., MipNeRF360, Tanks and Temples.

* Accessing cross-dataset generalization. It would be interesting to see how well the introduced CGA and IGR can generalize to other datasets. For example, trained on RealEstate10K but tested on DTU, similar to Fig. 5 in MVSplat.

* The motivation of using Deformable Attention is unclear. It might be better to provide experiments with further analysis to justify why it is better to use Deformable Attention instead of typical attention blocks.

* The paper title and model name are confusing. It might be better to highlight in the title that this work predicts an adaptive quantity of 3D Gaussians. Besides, the model name ‘PixelGaussian’ might unintentionally imply ‘pixel-align Gaussian’, which is the opposite of the objective. It might be better to change it to ‘AdaptiveGaussian’ or some other name that better reflects the main contribution of this work.

**Questions:**

Kindly refer to [Weaknesses].

---

> ### Author Response · Authors · 2024-11-25
>
> Thank you for your valuable comments and suggestions. Below we address specific questions.
>
> **1. Visualization of Depthmap and Point Cloud**
>
> We agree that additional visualizations are essential. We have included more comparisons of depth maps and point clouds, specifically using 4, 8, and 16 views as inputs in our revised paper. Thank you for your valuable suggestion.
>
> **2. More Input Views, Datasets and Cross-dataset Generalization**
>
> Thank you for the suggestion. We agree it is important to access more challenging view settings and perform cross-dataset generalization experiments. The results are listed in Table 1 and 2 in our general rebuttal. We have included these experiments in our revised paper.
>
> **3. Motivation of Deformable Attention**
>
> Thank you for pointing this out. Since the Gaussian predictions in our method are not strictly pixel-aligned, the projection of the Gaussian center may not precisely correspond to the specific location on the feature map. Therefore, we incorporate deformable attention into our CGA and IGR blocks, which enables more flexible and adaptive interactions to ensure each Gaussian can share information with the exact location in the score maps and feature maps. To further demonstrate our design, we conduct a simply ablation with 4 view inputs on the RealEstate10K dataset. When a rigid perception field is used for each Gaussian query in the CGA and IGR blocks, we observe an average performance drop of 1.58 in PSNR. A brief explanation of this issue have been added in our revised paper as you suggested.
>
> Table 1: PSNR comparison for ablations on deformable attention
> |       | rigid | deformable |
> |-------|-------|-------|
> | 4 views | 22.30 | 23.95 |
> | 8 views | 22.51 | 24.05 |
> | 16 views | 22.62 | 24.18 |
>
> **4. Paper Title**
>
> Thank you for the advice and we agree that PixelGaussian may lead to misunderstanding that our model predict pixel-aligned Gaussian representations. Therefore, we have changed the title to AdaptiveGaussian as you suggested.

---

### Author Response · Authors · 2024-11-25
**General Rebuttal**

We thank all the reviewers for their valuable feedbacks. In this general rebuttal session, we try to address the common questions for each reviewer and we will upload the revised paper shortly.

As requested by Reviewer bYDc, z7ms, and X5K2, we further evaluate our model under a broader range of input view settings on both the RealEstate10K and DTU datasets, to provide a more comprehensive assessment of its performance across different scenarios.

We evaluate our model, trained on the RealEstate10K dataset, for inference on both the RealEstate10K and DTU datasets. For each scene, we select the reference view to cover as wide a view range as possible to compare NVS results on such challenging cases. Due to memory limitations, we evaluate our model with a maximum of 16 input views and we split the inputs into batches to get Gaussians for pixelSplat on the 16-view setting.

Table 1: PSNR comparison of multiple input view settings on challenging cases in RealEstate10K dataset
|          | pixelSplat | MVSplat | PixelGaussian |
|----------|----------- |---------|---------------|
| 4 views | 22.02 | 22.30 | 23.95 |
| 8 views | 19.97 | 20.39 | 24.05 |
| 12 views | 18.92 | 19.69 | 24.18 |
| 16 views | 18.90 | 19.40 | 24.24 |

Table 2: PSNR comparison of multiple input view settings for cross-dataset generalization on DTU dataset
|          | pixelSplat | MVSplat | PixelGaussian |
|----------|------------|---------|---------------|
| 4 views | 12.30 | 12.45 | 13.42 |
| 8 views | 11.94 | 12.10 | 13.46 |
| 12 views | 11.75 | 11.78 | 13.53 |
| 16 views | 11.47 | 11.55 | 13.56 |

From the results, we observe that PixelGaussian slightly benefits from increasing input views in challenging cases while the performance of pixelSplat and MVSplat drops dramatically due to the overlap and redundancy of pixel-aligned Gaussians. Moreover, our model is able to maintain this advantage in cross-dataset generalization.

Please feel free to reach out if you have any follow-up questions or need further clarification. Your feedback is valuable to us, and we are happy to provide any additional information that may be helpful.

---

### Meta-Review · Area_Chair_1byz · 2024-12-17

**Metareview:**

This paper receives borderline ratings of 5,6,5,6. The AC look at the reviews, rebuttal and the discussions, and decide to reject the paper for now. The authors are encouraged to fix the issues and resubmit to future conference. The main concern raised by the reviewers is that the proposed method introduces extra overhead which lead to higher inference latency compared to MVSplat. Moreover, the performance is worse than MVSplat and rendering speed is similar to PixelGaussian when there are fewer than 4 views. These points suggest that the design is not sufficiently effective. Additionally, there are also concerns on the experiments such as more complex datasets, more input views, cross-generalization, etc. Although the two 6's are positive, there is no strong compelling reason to push for acceptance of the paper.

**Additional Comments On Reviewer Discussion:**

Although the discussions and rebuttal managed to get 2 reviewers to raise their scores, some concerns still remain from the other reviewers.

---

### Decision · Program_Chairs · 2025-01-22

Reject